# The Influence of Robot-Assisted Learning System on Health Literacy and Learning Perception

**DOI:** 10.3390/ijerph182111053

**Published:** 2021-10-21

**Authors:** Chun-Wang Wei, Hao-Yun Kao, Wen-Hsiung Wu, Chien-Yu Chen, Hsin-Pin Fu

**Affiliations:** 1Department of Healthcare Administration and Medical Informatics, Kaohsiung Medical University, Kaohsiung City 80708, Taiwan; cwwei@kmu.edu.tw (C.-W.W.); haoyun@kmu.edu.tw (H.-Y.K.); whwu@kmu.edu.tw (W.-H.W.); u108572003@kmu.edu.tw (C.-Y.C.); 2Department of Medical Research, Kaohsiung Medical University Hospital, Kaohsiung City 80708, Taiwan; 3The Master Program of Long-Term Care in Aging, Kaohsiung Medical University, Kaohsiung City 80708, Taiwan; 4Department of Marketing and Distribution Management, National Kaohsiung University of Science and Technology, Kaohsiung City 824005, Taiwan

**Keywords:** robot-assisted learning system, health education, health literacy, learning perception, the elderly

## Abstract

Healthy aging is a new challenge for the world. Therefore, health literacy education is a key issue in the current health care field. This research has developed a robot-assisted learning system to explore the possibility of significantly improving health literacy and learning perception through interaction with robots. In particular, this study adopted an experimental design, in which the experiment lasted for 90 min. A total of 60 participants over the age of 50 were randomly assigned to different learning modes. The RobotLS group learned by interacting with robots, while the VideoLS group watched health education videos on a tablet computer. The content dealt with hypertension related issues. This study used the European Health Literacy Survey Questionnaire (HLS-EU-Q16), Health Knowledge Questionnaire, Reduced Instructional Materials Motivation Survey (RIMMS), and Flow Scale as evaluation tools. The result shows no significant difference in the pre-test scores between the two groups. Compared with the video-assisted learning system, the robot-assisted learning system can significantly improve health knowledge, health literacy, learning motivation, and flow perception. According to the findings of this study, a robot-assisted learning system can be introduced in the future into homes and care institutions to enhance the health literacy of the elderly.

## 1. Introduction

With the increase of the elderly population and the average life expectancy, healthy aging is a new challenge facing the international community. People gaining better health knowledge and behaviors will help achieve the goal of healthy aging [1]. Health literacy education is one of the feasible solutions [2] and refers to the ability of individuals to obtain, process, understand and apply health information to make the right health decisions and take appropriate health actions [3,4]. Individuals’ inadequate health literacy may result in poor health decisions, behaviors, and outcomes [5,6]. Therefore, many countries regard health literacy education as an important national strategy [7].

When people encounter health problems, their most typical impulse would be to look for information through Internet search engines, ask relatives and friends, or watch health channels on TV. However, in general, they may not be able to judge the correctness of the information. Hence, if information and communication technology can play the role of providing correct health information, it will help fill the gap in health education and enable people to obtain adequate and correct health information and services. Furthermore, scholars consider that robots have great potential as a learning technology and can be applied to various types of educational purpose [8,9,10,11]. An educational robot has five basic educational applications, i.e., language education, robotics education, teaching assistance, social skill development, and special education, as well as guided learning through feedback [9]. Robots can serve as peers or mentors of learners and can effectively improve learners’ cognition and affection [12]. In addition, it is worth noting that traditional learning technology lacks the robot’s characteristic of physical presence [13]. The functions and interaction methods of robots are different from those of computers, tablets, and mobile phones. Most robots use facial expressions, body movements, and natural language to interact with users. Therefore, personalized learning content and methods can be designed in the robot to provide learners with a customized learning experience.

As elderly people spend longer at home, they may lack comprehensive and friendly health education channels. Meanwhile, it is possible to receive false health food advertisements on TV programs, causing great damage to their bodies. If there is a mechanism that can effectively improve the health literacy of the elderly, it will help maintain their physical and mental health, thereby reducing the medical burden on the family, society, and the country. Therefore, this study attempts to explore the feasibility and substantial utility of applying robotic technology to health education for the elderly. In particular, a robot-assisted learning system (RobotLS) based on humanoid robots was implemented to provide users with health information. The system interacts with users in a voice-based manner to reduce operational complexity. The findings of this study can contribute to home care and care institutions in improving people’s health literacy and well-being.

The health information received by the elderly mainly comes from newspaper advertising, radio, and television. Among these media, video is more effective in delivering knowledge. Therefore, this study uses videos as a benchmark for comparing the effectiveness of health education for the elderly. The ARCS (Attention, Relevance, Confidence, Satisfaction) motivation model proposed by Keller [14] aims to provide a systematic teaching evaluation model. If teachers design content based on the ARCS model, it will help learners engage in learning activities, thereby improving their motivation and effectiveness. Moreover, if the elderly can interact with the robot to generate a flow state, the RobotLS can help them focus and enjoy the interactive learning process. They can immerse themselves in the present and forget the passage of time. Therefore, this study uses robots for health literacy education, which is expected to help the elderly improve their learning motivation and flow state.

According to the above research background and purpose, this study proposes the following research question. Compared with the video-assisted learning system (VideoLS), can the robot-assisted learning system (RobotLS) significantly improve the health knowledge, health literacy, learning motivation, and flow perception of the elderly? In order to answer the research question, an experimentation method is used to evaluate the effectiveness and feasibility of robots in health education for the elderly. Chi-square and independent-sample t-tests are used to test whether there are significant differences in the demographic variables of the two groups. The paired sample t test is used to test whether the post-test scores of each group are significantly better than the pre-test scores in terms of health knowledge. Analysis of covariance (ANCOVA) is used to test whether there is a significant difference in the effectiveness of health knowledge learning between the two groups. Regression analysis is used to explore the effects of pre-test scores, learning modes, and demographic variables on post-test scores. Independent sample t-test is also used to analyze the differences in health literacy, learning motivation, and flow perception between the two groups. All analysis procedures are completed using IBM SPSS Statistics 26.

## 2. Health Literacy and Robot Application

### 2.1. Health Literacy

Health literacy is a multidimensional concept that involves a series of skills that people need to apply effectively and efficiently in a healthcare environment. Many studies have confirmed that there is a significant direct relationship between health literacy and health behaviors or health outcomes [15]. Insufficient health literacy can lead to poor health behaviors, disease prevention and treatment capabilities [16]. Therefore, health literacy is usually used as an indicator to measure the effectiveness of health education. Health literacy can be distinguished into three dimensions: functional, interactive and critical. Functional health literacy concerns the basic reading and writing skills for understanding and using health information. Interactive health literacy includes the advanced cognitive and literacy skills to be able to interact with healthcare providers, and the ability to interpret and apply information in a constantly changing environment. Critical health literacy is an advanced cognitive skill that can judge and analyze a variety of health information, and effectively apply judgment to improve one’s life [17,18].

Sørensen et al. [19] used a systematic review to analyze articles related to health literacy and proposed an integrated conceptual model. The process of people acquiring health literacy can be divided into four steps: access, understand, appraise, and apply [19]. If people can perform the following four procedures, it will help promote their personal health.

Access. The ability to find, discover, and obtain health information.Understand. The ability to comprehend the health information obtained.Appraise. The ability to interpret, filter, judge, and evaluate the obtained health information.Apply. The ability to communicate with professional medical staff and use health information correctly to make decisions to promote health.

Health literacy is a lifelong learning process. The knowledge acquired at all stages of life will be forgotten over time, and health knowledge may need to be updated. Therefore, people must have good health literacy to learn appropriate knowledge relevant to themselves. Sørensen et al. [19] divided the health issues people face in life into three domains: healthcare, disease prevention, and health promotion. Although the procedures for acquiring health knowledge in each domain are the same, the scope of application is different. Nevertheless, good health literacy helps people maintain and improve their quality of life.

Healthcare refers to patients who have fallen ill or are being cared for in a medical institution. It emphasizes the acquisition, understanding, and evaluation of medical or clinical information and the ability to make wise treatment decisions and comply with medical advice.Disease prevention refers to the state of being at risk of disease, emphasizing the retrieval of information on health risk factors, understanding, interpreting and evaluating relevant information, and making smart decisions that help maintain health.Health promotion refers to health education for the general public, including the efforts of communities, workplaces, education systems, political organizations, and the health industry to strengthen people’s health. It emphasizes regular updates of health information regarding the social and physical environment, leading to wise decisions.

Education is one of the important ways to improve people’s health knowledge and literacy. This study uses four steps—health information acquisition, understanding, evaluation, and application—as the basic framework for developing the RobotLS to promote learning among the elderly regarding health knowledge and literacy.

### 2.2. Robots for Healthcare and Education

In response to the trend towards an aging population and the shortage of nursing manpower, robots have gradually been used in home care or medical institutions. Since the elderly may easily feel lonely when living alone at home, manufacturers have also developed various types of companion or service robots. For example, the Robear developed by Riken of Japan can assist in carrying patients [20], and the RoNA developed by Hstar Technologies in the United States is a typical care-based robot that can assist medical practitioners in taking care of patients [20]. Furthermore, the Paro robot has the appearance of a seal and is usually used for life companions. When the user touches Paro, it will give appropriate feedback. The scholars apply Paro to accompany elderly people with dementia [21]. Another robot, Nao, is mainly based on dialogue interaction and body movements, and can be used as objects for users to talk and listen to [22].

At present, the application of robots in health education is mostly in special education. Scholars, such as van den Heuvel et al. [23] used the ZORA robot to intervene in the rehabilitation and special education of children with severe physical disabilities. A total of 17 disabled children, aged between 2 to 8 years old and seven professionals participated in the exploratory study over a period of 2.5 months. All participants participated in six activities related to the ZORA robot, including sports exercises, dance exercises, robot control, and cognitive exercises. The study results found that ZORA has greatly contributed to the rehabilitation treatment and cognitive development of children with disabilities, while children experienced a great deal of fun. Scholars consider that the robot is most suitable in three fields—motor, communication, and cognitive skills [19,20,21]. In addition, ZORA also helps stimulate learning motivation, concentration, an active attitude and an improvement in children’s attention. Conti et al. [24] used the Unified Theory of Acceptance and Use of Technology (UTAUT) to explore the important factors in using robots as special teaching tools. The overall results show that subjects have a positive attitude towards the use of robots. Since humanoid robots are often expensive, some people think that applying robots is impractical for educational purposes. However, research suggests that, as long as the benefits exceed the costs, robots still have considerable potential in the field of special education.

Although most studies support the benefits of robots in education, some scholars have given warnings about their application in special situations. Alcorn et al. [25] invited 31 autism educators (teachers, teaching assistants, and speaking and speech therapists) to discuss the role of social robots in the education of autistic learners through semi-structured interviews and focus groups. Although almost all interviewees are happy to use humanoid robots in the classroom (e.g., NAO, KASPAR, Milo), the paradox is that these educators worry that time will bring counter-effects. For example, robots can stimulate students’ learning motivation and classroom participation, but they may also hinder students from participating in group activities. Whether students with autism can transfer the knowledge or skills learned from the robot to the real environment, what are the substantial benefits and negative effects, and which functions of the robot can meet the specific needs of these students, all need more evidence and research.

From the above discussions, the nursing staff provided positive feedback on the robot used for assisted care. Robot-assisted education is often used in children’s rehabilitation and special health education and is found to improve the learning effectiveness of children and special needs students. However, there is no research on health education using robots for the elderly. Thus, it is unknown whether the application of robots to health education for the elderly can also produce positive effects. This study focuses on the elderly and robot health education to explore the effects of robot-assisted learning for this group.

## 3. Research Method

The framework of the health literacy education proposed in this study includes robots, cloud databases, and interactive algorithms. The user’s personal data and physiological information will be recorded in the cloud database, and the teaching materials placed in the cloud will be updated in a timely fashion. The system will provide appropriate health information and health improvement suggestions according to users’ needs. As a robot is mainly used as an interface for human–computer interaction, this study terms this system a robot-assisted learning system (RobotLS).

This study has designed two ways of presenting learning materials: a robot-assisted learning system (RobotLS) and a video-assisted learning system (VideoLS). The RobotLS uses a humanoid robot as a learning device to assist the elderly in learning health knowledge through voice and touch interactions, while the VideoLS uses animation videos to deliver learning contents. This study aims to explore the effects of the two learning modes on the elderly’s health knowledge, health literacy, learning motivation, and flow perception.

### 3.1. Learning System Development

In order to avoid the influence of exogenous variables on the research results, this study uses an experimental research method to explore the impact of different learning modes on the learning outcomes of participants. The experimental research method refers to the researcher’s investigation as to whether there is a causal relationship between the independent variable and the dependent variable under the control of irrelevant variables [26]. Therefore, during the experiment, subjects will be randomly assigned to the experimental group or the control group. Different experimental treatments will be then applied to different groups. The researcher can observe the influence of independent variables on dependent variables and explore its causality. The independent variables of this study are different learning methods, while the dependent variables are health knowledge, health literacy, learning motivation, and flow perception.

In this study, a humanoid robot named Robelf, as shown in Figure 1, was selected for system development. This robot has a pleasing appearance and uses a tablet to present facial features. When the robot interacts with the user, it can express happiness, anger, sadness, joy, and boredom. In addition, the robot also has basic functions, such as face recognition, remote control, information provision, real-time reminders, facilities for children’s education, home safety intelligent monitoring, and so on.

This study uses Android Studio and Robelf SDK to design a RobotLS. Since the subjects are elderly, the fonts are enlarged, and the color contrast is enhanced when designing the user interface. The learning content is mainly presented with keywords and images, and is explained by voice. The interactive design will slow down the speech speed, amplify the volume, and repeat reminders of key content [27]. Scholars [28,29] have pointed out that the behavior of the robot can make the user produce corresponding emotions and improve human–computer interaction. Body movements and hand postures can increase the robot’s popularity, anthropomorphism, and social participation. Facial expressions can improve users’ understanding of the interaction context and trust in the robot. Therefore, the RobotLS designed by this study can display facial expressions and body movements based on content and interaction status to enhance the sense of participation and identity from the elderly.

Many elderly people suffer from at least one chronic disease. The main types of common chronic diseases are cardiovascular diseases, such as hypertension, diabetes, hyperlipidemia, and heart disease. Among them, high blood pressure is an invisible killer. The control of high blood pressure is closely related to daily habits, exercise status, and diet. Therefore, this study uses hypertension prevention as the content of the experimental course. Consequently, the elderly, regardless of whether they have been diagnosed with hypertension or not, can learn about hypertension.

The experimental contents of this study are designed based on the hypertension prevention and treatment manuals published by government health agencies. It has two topics: knowledge of hypertension prevention and treatment, and diet control for hypertension. The related knowledge of hypertension prevention and treatment includes five units: introduction of blood pressure, introduction of hypertension, blood pressure monitor, medication, and lifestyle improvement. The diet control for hypertension includes three units: dietary style, diet seasoning, and diet choice. All learning materials have been confirmed by an attending physician in the family medicine department and two nurses to confirm its correctness. Experts in these fields have more than 10 years of clinical qualifications. Before the experiment, participants’ blood pressures were measured, and the experimental system gives the participants corresponding content based on their respective blood pressure statuses.

The RobotLS will follow the process of acquiring health literacy for interactive teaching in each unit. In the access stage, the robot actively provides users with health information based on personal blood pressure. In the understand stage, the robot will explain and use pictures, tables, animations, videos, and multimedia presentations to help users further understand the content. In the appraise stage, the robot will use the interactive method of question-and-answer to evaluate the user’s understanding of information and judgment of health events. Meanwhile, in the apply stage, the robot will ask questions on real-life situations to determine whether participants can apply health knowledge about their daily lives.

The content of the VideoLS is the same as that of the RobotLS to avoid the influence of content differences on the research results. Therefore, the production method of the VideoLS is to remove the robot’s question-and-answer and physical interaction capabilities. The content remains unchanged and retains the function of voice output. When learning, users can control the speed and progress of the presentation of the content to meet their learning situation. The VideoLS is shown in Figure 2.

### 3.2. Instrument

#### 3.2.1. Health Knowledge Test

There are 20 questions in the health knowledge test, with a full score of 100 points. Among them, 10 questions are on hypertension-related knowledge, and the other 10 questions are on precautions for hypertension diet. The questions are all true-false items, and the participants will need to mark “correct”, “wrong”, or “don’t know” based on the description. This study asked three clinical medical experts to confirm the appropriateness and correctness of these questions.

#### 3.2.2. Health Literacy Scale

In order to accurately measure people’s comprehensive health literacy, Sørensen et al. [30] designed and developed the European Health Literacy Survey Questionnaire (HLS-EU-Q). Since 47 items are excessively time-consuming, scholars have simplified the HLS-EU-Q47 into 16, 12, and 6 item versions, which are convenient for testing under the condition that the reliability and validity can be ensured [31]. This study uses the HLS-EU-Q16 version to measure the health literacy of the participants. HLS-EU-Q16 has a total of 16 questions, evaluating health literacy in health care (seven items), disease prevention (five items), and health promotion (four items). The scale uses a four-point Likert scale, ranging from “1 = very difficult” to “4 = quite easy”. According to the scoring formula proposed by Sørensen et al. [31], the health literacy score ranges from 0 to 50 points. Limited health literacy is 0–33 points; sufficient health literacy is 33–42 points; and excellent health literacy is 42–50 points.

#### 3.2.3. Learning Motivation Scale

Compared with the traditional teacher-oriented mode, the appropriate use of robots for teaching aids in the teaching process can help learners improve their learning interest, attitude, motivation, and effectiveness [32,33]. The four steps of the ARCS motivation model [14] include:Attention. The course design can arouse and maintain students’ curiosity and interest in the learning content.Relevance. The course design must meet students’ personal learning goals to promote positive learning attitudes.Confidence. The course design can help students construct self-confidence and complete learning tasks.Satisfaction. When students successfully achieve the preceding three goals, they can obtain internal and external encouragement and feedback, motivating them to continue learning.

The assessment of learning motivation uses the Reduced Instructional Materials Motivation Survey (RIMMS) developed by Loorbach et al. [34]. This scale is a condensed version of the motivation scale developed by Keller [35] from 36 to 12 questions. The test result reliability test of the overall questionnaire is better than the original scale. The purpose of using the RIMMS is to assess the influence of the learning mode on the learning motivation of the elderly. The scale is measured using a five-point Likert scale, from “1 = strongly disagree” to “5 = strongly agree”.

#### 3.2.4. Flow Perception

Csikszentmihalyi [36] proposed the concept of flow, which means that when people are engaged in an activity, they are fully immersed and attentive, and enjoy their current state of mind. Flow state can be interpreted as the best type of activity experience. In the flow state, people will be completely immersed in the current situation and ignore the perception of the outside world. Challenges and skills are two important factors that affect the perception of flow [37]. When challenges and skills reach a balance, a state of flow will occur. However, if people lack skills in facing great challenges, they will feel anxious about the current situation. In addition, when people have excellent skills but not enough challenges, they may get bored.

Webster et al. [38] pointed out that the higher the level of flow, the more autonomously they can participate in current activities. For different learners, learning tasks of appropriate difficulty must be designed to help learners immerse themselves in the learning process. The flow state can be evaluated using four dimensions, namely control, attention focus, curiosity, and intrinsic interest [36,38,39,40].

Control. Individuals have the ability to control the current situation or properly handle the operation of equipment.Attention Focus. The individual’s attention is focused on the current situation, showing a state of non-distraction. The focus of the flow state emphasizes the concentration of the interaction between the user and the device.Curiosity. The individual feels that the task being performed is novel and therefore has the intention to explore.Intrinsic interest. The individual feels fun in the current state and enjoys the process.

This study uses the flow scale developed by Webster et al. [38] to evaluate the flow perception of the elderly when using RobotLS or VideoLS. This scale has a total of 12 items, using a five-point Likert scale, ranging from “1 = strongly disagree” to “5 = strongly agree”.

### 3.3. Experimental Design

The participants of this study are healthy or sub-healthy elderly people who are over 50 years old and have retired. They have the ability to live independently and do not need the care of others. They can also listen, speak, read, and write Chinese. Participants were randomly assigned to the RobotLS and VideoLS groups. To avoid being overloaded, this study takes the critical issue of hypertension for the elderly as the health education content.

71 seniors were recruited to participate in the experiment—36 in the RobotLS group and 35 in the VideoLS group. For the RobotLS group, four elders thought that the interaction time with the robot was too long and interrupted the experiment, while two elders thought there were too many questions in the questionnaire and were unwilling to answer all of them. Among the participants in the VideoLS group, two elders interrupted the experiment, without the patience to watch the full video, two elders did not complete the questionnaire, and one elder had poor Chinese communication skills and could not understand the content of the video. Finally, there were 30 people in each group to complete the experiment.

Before the experiment, participants signed informed consent for the study and their blood pressure was measured and recorded. Then, they supplied personal information and finished a pre-test on health knowledge. After the experiment, a post-test of health knowledge was carried out, and the health literacy, learning motivation, and flow perception scales were filled in. In order to help participants to use the learning systems smoothly, this study instructed them in detail on how to use robots and tablets before conducting the experiment, including volume adjustment, progress control, functional operations, and interactive methods.

The intervention of the RobotLS group is designed to allow participants to interact with Robelf humanoid robots. The robot guides participants to learn about hypertension prevention and proper diet. Before the formal experiment, there were 10 minutes of robot operation instructions and interactive exercises so that participants could master the timing and method of answering the robot’s questions. Participants can decide whether to take a break at the end of each topic to avoid being overworked. The intervention of the VideoLS group was conducted by watching videos on the same health content through a 10-inch tablet computer. The system has play, pause, playback, and speed adjustment functions, allowing participants to control their progress in watching the video independently. The total experiment time was about 90 min.

## 4. Results 

### 4.1. Demographic Variables

The participant’s personal information collected in this study includes gender, age, education level, presence or absence of hypertension, exercise frequency (hours/week), and health class attendance frequency (times/week), as shown in Table 1. A chi-square test analysis showed no significant differences in categories, such as gender, education level, and hypertension. In addition, the independent sample t-rest showed no significant difference in continuous variables, such as age, exercise frequency, and health class attendance frequency. The result shows that this study has indeed achieved the purpose of random assignment.

### 4.2. Health Knowledge

As shown in Table 2, the mean of the pre-test on health knowledge of the RobotLS group was 55.50 (SD = 10.53), while that of the VideoLS group was 55.83 (SD = 10.51). After participating in experiments with different learning modes, the mean of the post-test of the RobotLS group was 85.33 (SD = 9.91), while that of the VideoLS group was 75.50 (SD = 11.32). The paired sample t-test results showed that the health knowledge of the two groups was significantly improved. In other words, regardless of which learning mode was adopted, participants can effectively learn from well-planned and designed learning materials.

Since the participants’ health knowledge was inconsistent before the experiment, this study used an analysis of covariance (ANCOVA) to evaluate the impact of the learning mode on post-test health knowledge in order to exclude the influence of prior knowledge on learning effectiveness. In this study, the pre-test health knowledge was used as a covariate. Although the pre-test scores of the participants had a significant impact on the learning, there was still a significant difference in the post-test health knowledge between the two groups (F = 19.423, *p* < 0.001) as shown in Table 3. This study verifies that the learning effectiveness of the RobotLS group was significantly better than that of the VideoLS group.

In order to understand whether demographic variables will also have a significant impact on learning effectiveness, regression analysis was used to explore the effects of pre-test scores, learning modes, and demographic variables on post-test scores. The analysis results showed that the variables that have a significant impact on post-test scores are learning modes and pre-test scores, as shown in Table 4. This finding is consistent with the result of ANCOVA analysis. That is, the participants’ prior knowledge of hypertension and the way they interact with the learning materials have significant influence on the participants’ learning effectiveness.

### 4.3. Health Literacy

In this study, the HLS-EU-Q16 scale [41] was used to evaluate participants’ health literacy. Health literacy is composed of three constructs: health care, disease prevention, and health promotion. The Cronbach’s α are 0.884, 0.872, and 0.887, respectively, which shows that the health literacy scale used in this study has a high degree of internal consistency.

Table 5 shows the health literacy scores of the participants measured after the end of the experiment. The mean of the health literacy in the RobotLS group was at a good level, while that in the VideoLS group was at a sufficient level. In all the health literacy constructs, the RobotLS group is significantly better than the VideoLS group. The results show that the participants’ health literacy can reach an ideal level through an appropriate learning process. Moreover, the learning performance will be more significantly improved with the support of human–robot interaction design.

### 4.4. Learning Motivation

This study used the RIMMS [34] to measure participants’ learning motivation. The scale comprises four dimensions: attention, relevance, confidence, and satisfaction. The Cronbach’s α are 0.924, 0.905, 0.912, and 0.919, respectively, indicating the questionnaire’s good internal consistency reliability.

According to the statistical analysis in Table 6, the means of the RobotLS group in the four dimensions of learning motivation are significantly better than those of the VideoLS group. This shows that the involvement of robots in health education helps arouse users’ motivation for learning.

### 4.5. Flow Perception

The flow scale is composed of four dimensions: control, attention focus, curiosity, and intrinsic interest [38]. The Cronbach’s α are 0.874, 0.858, 0.935, and 0.893, respectively, indicating the questionnaire’s good internal consistency reliability.

Table 7 shows the mean, standard deviation, and test results of the two groups in terms of flow perception. The mean of each dimension of the RobotLS group is significantly better than that of the VideoLS group. This shows that when users use robots to learn, they can immerse themselves in the learning situation and arouse their curiosity and intrinsic interest in participating in the course.

## 5. Discussion

The results show that the health knowledge of the RobotLS group is significantly better than that of the VideoLS group. In the process of human aging, memory tends to gradually decline with age. The elderly may also be unable to concentrate for a long time, which may reduce the effectiveness of learning new content. Their memory can be improved and maintained through repetitive exercises and the use of appropriate strategies. Liu et al. [42] pointed out that asking the elderly to answer relevant questions after learning and then providing appropriate materials based on the correctness of the answers will help improve their learning effectiveness. The RobotLS developed in this study has a question-and-answer function, which can immediately assist in reviewing correct knowledge when the elderly make incorrect answers. The system will increase the interest and pleasure of learning for the elderly through appropriate interaction, rather than just the transfer of knowledge.

Humanoid robots can make users feel their physical and social presence [13,43]. The robot used in this study has a human-shaped structure and is capable of simple dialogue and physical interaction with the user. The robot is given an identity symbol. It appears as a cute boy at the age of 8, and his name is Xiaobei. When conducting the experiment, Xiaobei would introduce himself and greet the elders, just like a young grandson naturally sharing health knowledge with his grandparents. The research finding confirms that a good human–robot interaction process is indeed helpful for the acquiring of health knowledge. Some studies have found that when people get along with a specific doll or robot for a certain period, they will regard it as a listener or companion [44,45]. Additionally, they will also have emotional engagement. This phenomenon is especially obvious for preschool children and elderly people because they lack the stimulation of new events. Therefore, compared with traditional learning methods, the RobotLS can improve the learning performance of the elderly more effectively.

The result shows that the health literacy of the RobotLS group was significantly higher than that of the VideoLS group. The average scores of the RobotLS group fall within the standard of good health literacy (42–50 points), and those of the VideoLS group falls within the interval of sufficient health literacy (33–42 points). A high health literacy score means that the user has a more precise concept of access, understanding, evaluation, and application of health information. The most common health problems faced by the elderly are whether or not to believe in drug or health food advertisements, dealing the health experience of relatives or friends, and online health information. The robot in this study is only used as a front-end device for interacting with users and presenting health materials. The complete RobotLS also has a back-end cloud database that records personal health status, medication usage, medical needs and other information. The system can provide appropriate and correct health information according to the user’s physical condition to prevent the elderly from misbelieving the wrong health information. The RobotLS in this study considers the characteristics of the elderly’s health information needs and is designed and developed based on four important procedures for health literacy. This is also one of the reasons why the health literacy of the RobotLS group is better than that of the VideoLS group.

The results show that the RobotLS group’s learning motivation in the four dimensions—attention, relevance, confidence, and satisfaction—is significantly better than that of the VideoLS group. The anthropomorphic characteristics and interactive design of robots are the main factors affecting learning motivation. Notably, similar findings have been found in previous research. Liew et al. [46] pointed out that the enthusiastic teaching model of virtual teachers can significantly enhance learning motivation and students’ willingness to use multimedia materials. Hsieh [33] considers that, compared with the traditional teacher-centered model, adding robot-assisted teaching can effectively increase the motivation of students to continue learning.

Previous studies have pointed out that when learners feel a high degree of flow in the learning process, it will help improve learning performance [40]. The results show that the RobotLS group’s control, attention focus, curiosity, and inner interest in the flow perception are also significantly better than the VideoLS group. Skadberg and Kimmel [47] pointed out that attractiveness is an important factor in the flow state, while robots also have the characteristics of attracting the attention of the elderly. The RobotLS has anthropomorphic characteristics and interacts with participants with voice, facial expressions, lights, and body movements. Webster et al. [38] indicate that people tend to engage in activities that make them happy. Therefore, the higher the degree of flow perception, the more one can encourage learners to participate in the learning process. This is also one of the key factors in improving learning performance and health literacy.

In constructs of learning motivation and flow perception, the significant level of confidence and control is smaller than that of other variables. Confidence is a measure of the participants’ confidence in operating the device to complete the learning task, while control is a measure of the participant’s ability to actually operate the learning device. For participants, RobotLS is definitely a relatively novel learning system. Although participants were asked to practice RobotLS before the experiment, they sometimes still needed the help of researchers in actual operation. Therefore, the means of confidence and control in the RobotLS group are the lowest scores in the construct of learning motivation and flow perception, respectively. Since participants may use mobile phones or computers to access information and engage in social activities, they may not find it difficult to use a tablet for learning. For the participants in the VideoLS group, the mean of the confidence variable is higher than the other three variables in the learning motivation construct, while the mean of the control variable is higher than the attention focus and curiosity variables in the flow perception construct. Due to the differences in the participants’ familiarity with learning equipment, the two groups are less significant in the two variables of confidence and control. Based on the findings of this research, we suggest that future research should give older people longer practice time to adapt to the use of new learning technologies.

## 6. Conclusions

In the past, the application of robots in education mainly focused on formal school education and special education. While people need more health knowledge as they grow older, such demands are often ignored. This study integrates the four core procedures of health literacy into the design of the RobotLS to deliver health knowledge. This study verifies the feasibility and effectiveness of this learning method with experimental research. The result shows that, compared with traditional video-based health education, the RobotLS helps improve the health knowledge, health literacy, learning motivation, and flow perception of the elderly.

There are some research limitations that must be considered when applying the results. Participants in this study are members of senior-age institutions. Because the members of these institutions are mostly women, female participants are obviously higher than male participants, thus showing that older women are more willing to participate in community group activities and leisure courses. Future research can invite male subjects extensively to understand how male elderly people feel about using the RobotLS. Moreover, it is difficult recruit a large number of elderly participants. Although the total sample is 60, each test has reached a significant level, indicating the feasibility of using robots for health literacy education. In the future, researchers can expand the number of samples on the basis of our research to increase the power of the test.

At present, the robot can only speak Chinese and recognize Chinese pronunciation, and there is no corpus built for other national languages and local dialects. The elderly who can only communicate in local dialects are more disadvantaged in obtaining correct health information. It is hoped that there will be a complete language package in the future so that the RobotLS can reach more users and exert greater utility.

This study is limited by the amount of experimental equipment and the difficulty encountered in long-term experiments. It also takes into account the physical strength, endurance, and concentration of the elderly. Therefore, the experiment time was set to 90 min. Although it is hard to ask participant to repeat experimental interventions, the research results still show that the RobotLS can achieve significant learning performance. If it can be implemented for a longer period, it is expected to continuously improve health knowledge and literacy.

Presently, the RobotLS takes the four procedures in obtaining health literacy as the main design. In the future, we can further discuss the design methods and application effects of robots in the three major areas, namely health care, disease prevention, and health promotion.

## Figures and Tables

**Figure 1 ijerph-18-11053-f001:**
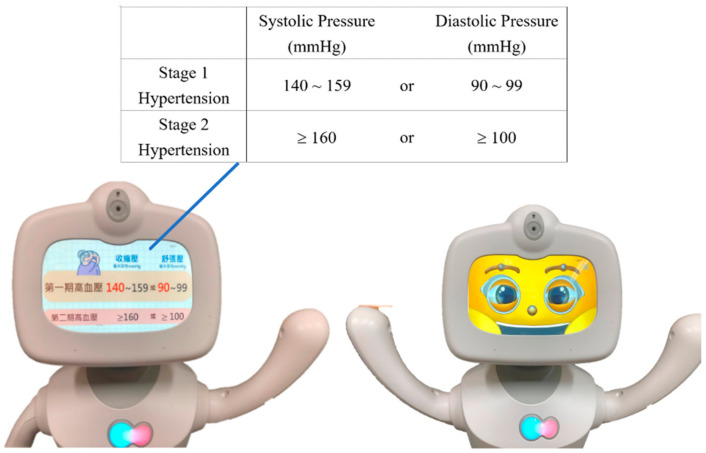
The robot-assisted learning system.

**Figure 2 ijerph-18-11053-f002:**
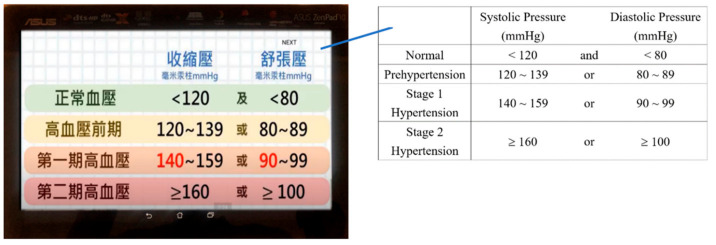
The video-assisted learning system.

**Table 1 ijerph-18-11053-t001:** Results of demographic variables.

Variable	RobotLS Group (*n* = 30)	VideoLS Group (*n* = 30)	*p*-Value
Gender			1.000
Male	8 (26.7%)	7 (23.3%)	
Female	22 (73.3%)	23 (76.7%)	
Age	70.87 ± 8.35	70.60 ± 7.57	0.897
Education level			0.952
Junior high school and below	13 (43.3%)	12 (40.0%)	
High school	9 (30.6%)	9 (30.0%)	
University	8 (26.7%)	9 (30.0%)	
Hypertension			1.000
Presence	16 (53.3%)	15 (50.0%)	
Absence	14 (46.7%)	15 (50.0%)	
Exercise frequency	3.80 ± 1.40	3.57 ± 1.59	0.549
Health class attendance frequency	2.63 ± 1.35	2.30 ± 1.06	0.291

**Table 2 ijerph-18-11053-t002:** Paired sample t-test of pre- and post-test regarding health knowledge.

Group	*n*	Pre-Test		Post-Test		t	*p*-Value
		Mean	S.D.	Mean	S.D.		
RobotLS	30	55.50	10.53	85.33	9.91	14.419	<0.001
VideoLS	30	55.83	10.51	75.50	11.32	13.499	<0.001

**Table 3 ijerph-18-11053-t003:** The result of ANCOVA on health knowledge. Dependent variable: post-test on health knowledge.

Source	Type III Sum of Squares	df	Mean Square	F	*p*-Value	Partial Eta Squared
Corrected model	3590.913 ^a^	2	1795.457	23.135	<0.001	0.448
Intercept	4668.831	1	4668.831	60.159	<0.001	0.513
Pre-test	2140.497	1	2140.497	27.581	<0.001	0.326
Learning mode	1507.353	1	1507.353	19.423	<0.001	0.254
Error	4423.670	57	77.608			
Total	396,025.000	60				

^a^ R Squared = 0.448 (Adjusted R Squared = 0.429).

**Table 4 ijerph-18-11053-t004:** The result of regression analysis on health knowledge. Dependent variable: post-test on health knowledge.

Source	Unstandardized Coefficients		Standardized Coefficients Beta	t	*p*-Value	VIF
	B	S.E.				
Constant	58.201 ^a^	12.279		4.740	<0.001	
Pre-test	0.375	0.131	0.335	2.863	0.006	1.629
Learning mode	9.491	2.174	0.411	4.366	<0.001	1.049
Age	−0.231	0.153	−0.157	−1.507	0.138	1.282
Gender	0.169	2.853	0.006	0.059	0.953	1.356
Education level	2.716	1.596	0.194	1.702	0.095	1.544
Hypertension	0.060	2.306	0.003	0.026	0.979	1.180
Exercise frequency	1.248	0.864	0.159	1.444	0.155	1.447
Health class attendance frequency	1.269	1.066	0.132	1.190	0.240	1.463

^a^ R Squared = 0.570 (Adjusted R Squared = 0.503).

**Table 5 ijerph-18-11053-t005:** Independent samples t-test on health literacy.

Construct	RobotLS Group (*n* = 30)		VideoLS Group (*n* = 30)		t	*p*-Value
	Mean	S.D.	Mean	S.D.		
Health care	44.84	5.42	33.89	6.15	7.320	<0.001
Disease prevention	44.67	5.07	32.22	6.86	7.992	<0.001
Health promotion	44.58	7.10	33.61	8.68	5.358	<0.001
Health literacy	44.72	4.60	33.30	6.05	8.235	<0.001

**Table 6 ijerph-18-11053-t006:** Independent samples t-test on learning motivation.

Construct	RobotLS Group (*n* = 30)		VideoLS Group (*n* = 30)		t	*p*-Value
	Mean	S.D.	Mean	S.D.		
Attention	4.54	0.62	3.63	0.82	4.856	<0.001
Relevance	4.66	0.49	3.83	0.64	5.573	<0.001
Confidence	4.38	0.67	3.98	0.64	2.368	0.021
Satisfaction	4.64	0.51	3.80	0.67	5.493	<0.001

**Table 7 ijerph-18-11053-t007:** Independent samples t-test on flow perception.

Construct	RobotLS Group (*n* = 30)		VideoLS Group (*n* = 30)		t	*p*-Value
	Mean	S.D.	Mean	S.D.		
Control	4.20	0.70	3.72	0.55	2.950	0.005
Attention focus	4.50	0.60	3.63	0.77	4.829	<0.001
Curiosity	4.38	0.69	3.42	0.88	4.659	<0.001
Intrinsic interest	4.54	0.48	3.87	0.53	5.217	<0.001

## Data Availability

Data are available upon reasonable request by contacting cwwei@kmu.edu.tw. The data are not publicly available due to privacy concerns.

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
