# Peer review of "The Influence of Robot-Assisted Learning System on Health Literacy and Learning Perception"

_ijerph, 2021, doi:10.3390/ijerph182111053_

Round 1
Reviewer 1 Report
This study developed a robot-assisted learning system to explore the possibility of significantly improving health literacy and learning perception
through interaction with robots. The results looks very interesting given some research limitations. I ask authors to revise the manuscript and address below comments/feedback:
1- In Research Method section, the statistical method and the software than was recruited should be addressed.
2- In line 398, covariance should be covariate.
3- This is not clear, how the sample size was obtained. Is it based on past studies or they have used some statistical formula based on an effect size?
4- It was mentioned that ANCOVA model has been used, but all tables 3, 4 and 5 contain p-values of two-independent T tests.
5- The right statistical method is ANCOVA and should be used here. I ask authors to incorporate group (video and robot), sex, age and the pre-test of health knowledge in an ANCOVA model. The new results should be reported in tables and the conclusion should be rewrite based on the correct model.
Author Response
Dear Reviewer:
Thanks for your valuable comments. Attached please the letter of replying your comments.
BRs
Hsin-Pin Fu

Reviewer 2 Report
- This topic is very interesting. I think this paper having more important contributions for academic researching community.
- In the section 2.1, I suggest the author could provide more theory about health literacy to fulfill the theoretical background.
- In the section 3, I suggest the author should provide the theoretical contents of the experimental design.
- In the section of results, the author provide appropriate statistical analysis about this topic.
- In the section of discussion and conclusion, I suggest the author could provide more comparative analysis about the differences between two groups. For example, to consider the reasons about the low significance of the construct of confidence in table 4 or control in table 5.
Author Response
Dear Reviewer:
Thanks for your valuable comments. Please find the letter of replying your comments.
BRs
Hsin-Pin Fu

Round 2
Reviewer 1 Report
Authors addressed my comments/suggestions in the current version of manuscript. I'm happy with the current version.